# "Once the Fire Starts Then There Is No Stopping It": The Revitalization of Chinookan Art in the 21st Century, Conversations with Greg A. Robinson

**Jon D. Daehnke** [1,*] **and Greg A. Robinson** [2,*]

1    Department of Anthropology, University of California, Santa Cruz, CA 95064, USA
2    Chinook Indian Nation, Bay Center, WA 98527, USA
*    Correspondence: jdaehnke@ucsc.edu (J.D.D.); palixchinook@gmail.com (G.A.R.)

**Abstract:** Chinookan art centered on the Lower Columbia River and was created by Chinookan-speaking people living along the river and its tributaries. The style is unique, focusing on geometric forms, numerical patterns, and anatomical representation. It is embedded in Chinookan mythology and differs considerably from the more widely recognized Formline of Indigenous artists from the northern Pacific Northwest. It also receives less attention, both publicly and scholarly. Due to high rates of death along the Columbia from introduced diseases during colonial invasion, and high levels of looting that followed, Chinookan art nearly disappeared from the landscape. In the 21st century Chinookan art has had a resurgence, led by Chinookan practitioners. The resurgence occurs not only within individual households but also in public settings. This resurgence also includes an emphasis on teaching the style to youth, who learn that this is not just about making art but is integrally attached to culture more broadly, including connection to language, stories, protocols, and Indigenous identity itself. It is ultimately a source of pride, resilience, and resistance. As a result, where there were once generations who never saw a landscape with Chinookan art, there are now generations who will never know a landscape without it.

**Keywords:** Chinook; art revitalization; indigenous art; northwest coast formline; public art; columbia river; indigenous resilience; colonialism

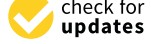



## 1. Introduction

On 8th June 2019, the *Forest of Dreams: Ainu and Native American Woodcarving* exhibition opened at the Portland Japanese Garden, in Portland, Oregon (Figure 1a,b). The exhibition was a commemoration of the 60th anniversary of the sister-city relationship that exists between Portland, Oregon, and Sapporo, Hokkaido in Japan. It showcased the artistry and traditions—especially through a focus on woodcarving—of Chinookans, who are the Indigenous people who have occupied the Lower Columbia River for millennia, and the Ainu, who are the Indigenous people of Japan. Monument-sized "Power Boards" were the centerpieces of the exhibit. For Chinookans, Power Boards were traditionally large paintings or carvings that could represent family identity, could be infused with a personal guardian spirit, and could serve as figures of welcome or spiritual protection. They were often found in houses, especially in the rear of the house where the chief resided. Chinookans view these carved Power Boards as living beings. Greg A. Robinson, a citizen of the Chinook Indian Nation, carved the Chinookan Power Board (Figure 2). In addition to these central pieces, *Forest of Dreams* also included other carvings by Greg Robinson, work from artist Tony A. Johnson of the Chinook Indian Nation, and artists Greg Archuleta and Bobby Mercier of the Confederated Tribes of Grand Ronde (Figure 3a,b).

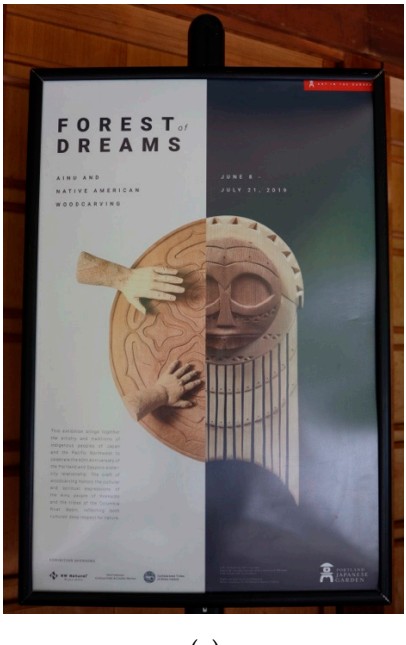

(**a**)

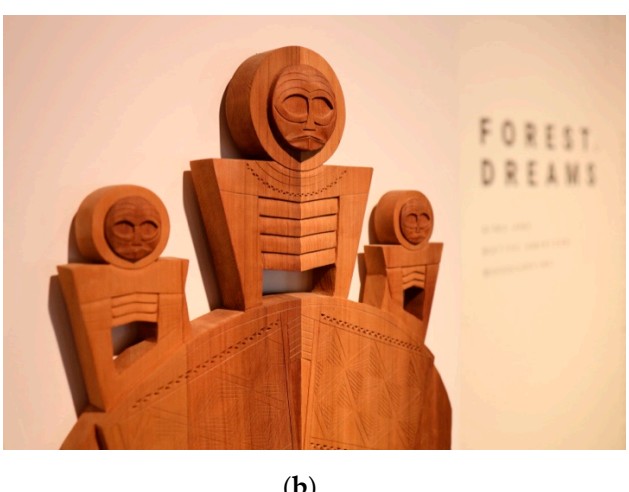

(**b**)

**Figure 1.** Signage for *Forest of Dreams: Ainu and Native American Woodcarving* exhibit at the Portland Japanese Garden, 8 June–1 July 2019. Photos by Jon Daehnke.

*Forest of Dreams* did more than celebrate the continuing relationship between Portland and Sapporo. Perhaps most importantly, the exhibit recognized and centered on Chinookan-style art, the art form indigenous to the lands on which Portland, Oregon, and the Japanese Garden now occupy. Chinookan art is a unique art form created by Chinookan-speaking people living along the Lower Columbia River, that stretch of the river that runs from present-day Dalles, Oregon, to the river's mouth, as well as along the Willamette River from Willamette Falls to its confluence with the Columbia. The style was also shared widely with the non-Chinookan neighbors with whom the Chinook interacted. While there are some regional variations in Chinookan art along the Lower Columbia and its tributaries, there are recognizable conventions and patterns that connect the style across geographic space. There is a consistent focus on geometric forms, numerical patterns, and unique representations of human and animal anatomy. The form is also typically embedded in Chinookan stories and mythology. Chinookan art appeared in both two-dimensional and three-dimensional forms, and a wide range of materials were utilized in its production. Columbia River basalt and pumice were often used for stone carvings, Western red cedar was a standard medium for wood carvings, although alder, maple, ash, and yew were

utilized as well. Bone and antlers were carved into figurines, adze handles, ladles, and bowls were crafted from mountain sheep horns (see Boyd 2018; Johnson and McIsaac 2013; Johnson 2014; Wingert 1952). Because of both their beauty and the technical craftsmanship that went into their creation, sheep horn bowls and ladles were especially coveted by both private and institutional collectors. Chinookan art differs considerably from the more widely recognized and ubiquitous Northwest Coast Formline work of Indigenous artists in the northern communities of the Pacific Northwest, as has been described and analyzed by Bill Holm ([1965] 2015). We will discuss the uniqueness of this art form later in the article.

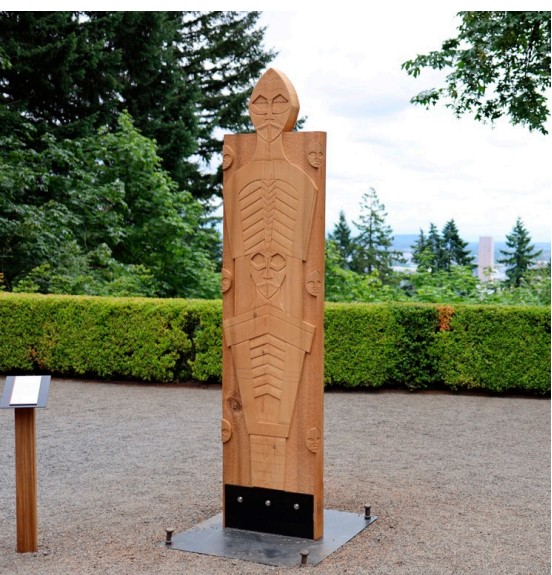

**Figure 2.** Power Board carving at the *Forest of Dreams* exhibit by Greg Robinson. Photo by Jon Daehnke.

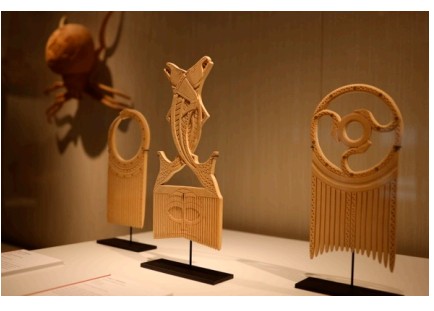

(**a**)

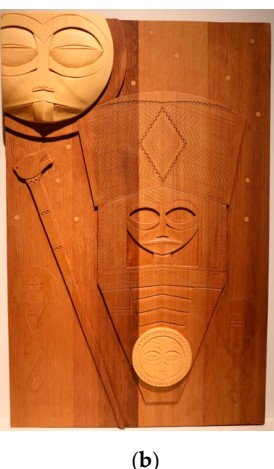

(**b**)

**Figure 3.** Carved combs (**a**) and carved panel (**b**) by Greg Robinson, at the *Forest of Dreams* exhibit. Photos by Jon Daehnke.

In the years following colonial invasion, the art form effectively disappeared from the landscape of the Columbia River. Introduced disease served as a primary cause of devastation. Smallpox may have reached the greater Pacific Northwest as early as the 1520s, and there is tangible evidence for smallpox epidemics occurring in the region in 1781-82, 1801-02, 1836-1838, 1852-53, and 1862. While smallpox was devastating to Indigenous populations throughout the Pacific Northwest, the worst epidemic to hit the region of the Columbia River—in terms of both casualties and the cultural disruptions that followed—began in the summer of 1830. Called the "fever and ague" by the Americans and the "intermittent fever" by the British, this epidemic—most likely malaria—raged through the lower Columbia and Willamette River valleys for several summers. The epidemic had a destructive effect on native populations. Both the Hudson Bay Company and Lewis and Clark had estimated a regional native population of somewhere near 15,500 in the early decades of the 1800s. By 1841, this number had been reduced to 1932, a decline of nearly 90 percent (see Boyd 1990, pp. 137–42; 1999, pp. 21–60, 160–201; 2013, pp. 235–38). Elders who held the physical knowledge of how to produce art, and who knew the stories and protocols so centrally necessary for its creation, were hit especially hard by introduced disease and colonial invasion. As a result, Chinookan art became essentially absent on the Lower Columbia River landscape where it was once ubiquitous. Furthermore, the Pacific Northwest Coast art that was publicly present in the region in the 20th century was most likely to be a type of geographically misplaced Northwest Coast Formline.

There has, however, been a dramatic resurgence in the creation of Chinookan art in the twenty-first century, and a broader public appreciation of it, as the well-attended *Forest of Dreams* exhibition demonstrated. The 2005 *People of the River: Native Arts of the Oregon Territory* exhibit at the Portland Arts Museum, sponsored by the Confederated Tribes of Grand Ronde, was the first major museum exhibit to focus on the historic art form of the Indigenous populations living along the Columbia River (see Mercer 2005). Without question, this exhibit of historic pieces led to greater initial recognition of the Chinookan art form by the general public. The primary driver of the form's present revitalization, however, has been the re-growth in the number of Chinookan carvers and artists in recent decades, principally in the communities of the Chinook Indian Nation and the Confederated Tribes of Grand Ronde. They are also now teaching the form to the next generation of carvers and artists. Furthermore, the revitalization of the form is not just present within the households of Chinookan practitioners and on tribal lands but is now again publicly present on the landscape where it properly belongs, as demonstrated in the *Forest of Dreams* exhibition. Chinookan public art can now be seen at the Cathlapotle Plankhouse at the Ridgefield National Wildlife Refuge, on the Portland State University Campus, at Parker's Landing Historical Park in Washougal, at Blue Lake Regional Park near Portland, in George Roger's Park in Lake Oswego, at the Oregon Convention Center, and anchoring both ends of the Tilikum Crossing Bridge in Portland, to provide a few examples. Chinookan art is also now available at galleries.

One of the central figures in the revitalization of Chinookan art is Greg A. Robinson, a citizen of the Chinook Indian Nation. The primary material for the remainder of this article comes from two recorded conversations between Greg and myself. Both of the conversations took place in Vancouver, Washington; the first on 7 August 2019, and the second on 4 April 2023. Each were roughly an hour and a half in length, and they addressed and revisited similar themes and questions. In these conversations Greg and I spoke at length about his introduction to the Chinookan art form, the characteristics of Chinookan art, the guides or information he used to help recreate and restore the form and the challenges in that, the inspiration for the resurgence of Chinookan art, some of the continued challenges associated with the lack of recognition of Chinookan art, especially in comparison to Northern Formline, and the importance of having Chinookan art present and visible on the landscape where it belongs. The following text is a condensed and combined version of those conversations. In addition to the two recorded conversations, our discussion in this article is informed by a more informal conversation that Greg and I had in January 2019

at the Grande Ronde Cultural Room at the Confederated Tribes of Grand Ronde Portland Office, during a culture education class. Finally, the background of this work is informed by roughly two decades of interactions, conversations, and collaborations between myself [Daehnke] and the Chinook Indian Nation (see Daehnke 2017). Greg Robinson and I have reviewed the following text for accuracy in terms of both detail and spirit.

## 2. Conversations

**Jon Daehnke**: To start, could you give a bit of a description of the Chinookan art style? What are some of the patterns that are seen within it? What differentiates it? What makes it unique to other artwork traditions in the Pacific Northwest?

**Greg Robinson**: Sure. So, luckily, the Chinookan form is unique enough that you're able to push away the other outside influences, such as Northern Formline and some of the Coast Salish forms that were more widely known when I started trying to put this art form back together. Here's where the Chinookan stories come into play with the art, because the stories have lines in them of repetitions of threes and fives. Triadic and quintic references, and that's throughout the stories. It's a very strong element. The art form, when you start looking for those patterns of repetition that I was looking for, there are also threes and fives everywhere (see Figures 4 and 7a,b). This wasn't a secret. It was not some big discovery that I made. But it became really evident that it was important in the art form. Those are the primary distinguishers, although it's not a rule written in stone. You see the rule broken occasionally because it's not always practical to put threes and fives in every design. But I think it was followed at every opportunity.

The next thing, besides a numerical reference, are the geometrics. The most prominent geometric is the triangle, and mostly in its ability to raise in the positive form—the triangle being the negative—raise the positive form of the zigzag (see Figure 5a,b). This is most evident on things like the sheephorn bowls. Also, some of the wooden bowls—the few that survived—and you see it on bone pieces as well. You don't see it as much on stone because it's more difficult to achieve a negative triangle on a stone piece. You also see concentrics, repeating concentrics, in all the forms—triangular, circular, rectangular, square, even just straight lines. You see those regularly.

There's also this X-ray effect, what's been labeled as an X-ray effect. It refers to elements that you might see from supernatural beings who are connected directly to the land of the dead. Most, I would say 90%, of the carvings or artifacts that we have access to show evidence of being tied to the land of the dead. And certainly, a hundred percent of it is tied to the supernatural world, but not all is necessarily tied to the land of the dead. The X-ray effect is a skeletonized corpse-like effect that references that, the power of that. And oftentimes you'll see the threes and fives correlated into that. Exaggerated collarbones, ribs, eyebrow ridges, and nose planes are above everything else on the face (see Figures 2, 4 and 6). This is also supported by the stories, where there was a constant mixture and no separation between the supernatural world and the existing world. It was all interwoven into itself. And that's the same for the art. The art was just a thread of that. One spoke of a weaving of the entire basket.

**JD:** What led you to Chinookan Art? Were you carving or doing other types of art as a youngster?

**GR:** So, I had a long career in construction, which kept me exhausted. I was doing some carving, mostly on gunstocks, because I was a pretty avid outdoorsman. But, I didn't get tuned into the Chinookan form right away. My dad and I were separated early on through divorce and I didn't have any contact with him, no real substantive contact. We reconnected twenty-some years ago. He's Chinook but enrolled with Quinault, which is a common story. And I have earlier memories of him fishing on the Columbia River. He had his blue card, which enabled him to fish on the Lower Columbia River. So, I have some of those memories, but not too much in the cultural sense. But when I reconnected with him, he got me involved with the Chinook Indian Nation and I started attending events. And then I joined the Culture Committee, and then I got on Tribal Council. I spent nine years on Tribal Council, and about ten years on the Culture Committee.

Once I was on the Culture Committee, we started meeting with the U.S. Fish and Wildlife Service, who were pitching the idea of a Chinookan Plankhouse at the Ridgefield National Wildlife Refuge to coincide with the Lewis and Clark Bicentennial (2003 to 2006). After many meetings, we were skeptical of their ability to build a legitimate Plankhouse, which is all we were interested in. But they eventually convinced us. I was probably the biggest skeptic because I'd been in the building trades and was aware of building codes, all of that, and didn't see how they could build that house on the refuge. They assured me that they could and that they could call it an "historic" building. Finally, I relented, and got involved in the project. And not too long after that, as their project moved forward, funding came into place (for further discussion on issues surrounding the Cathlapotle Plankhouse see Daehnke 2013, 2017).

They were looking for a cultural liaison project manager for the Plankhouse. I was looking for an escape from plastering at the time. So, I applied and got the job. The first six months or so was mostly planning and gathering logs and materials needed for the house. Once we started actually working on it we had a big volunteer crew that would come in on the weekends and work on fabricating the logs for the house. And at this point I was gaining interest in the art form [the Cathlapotle Plankhouse contains Chinookan carvings and paintings] and started doing carving. I wasn't allowed to do any of the commissioned carvings because of my position, but I was able to do a bunch of rudimentary textures and shapes, and things like that. So, that's really where the refocus on the art starts (Figure 4).

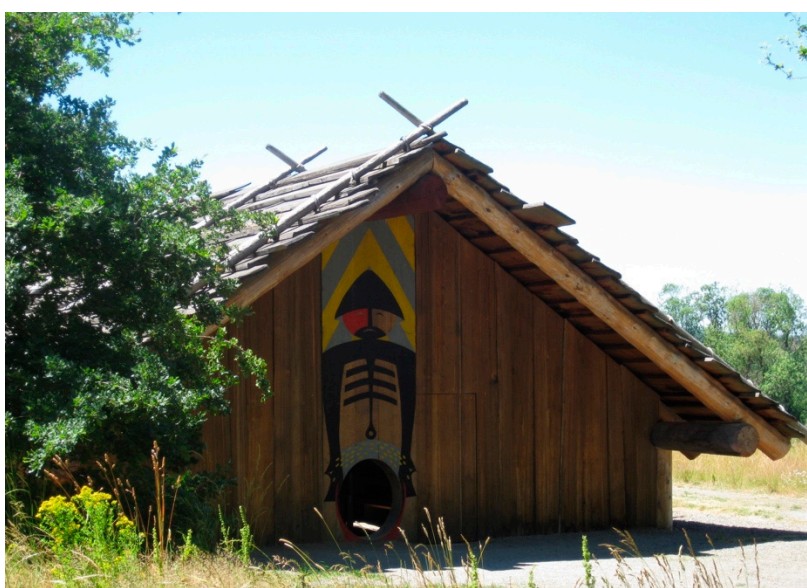

**Figure 4.** The Cathlapotle Plankhouse, Ridgefield National Wildlife Refuge, Ridgefield, Washington. In addition to the Chinookan design on the front of the plankhouse, Chinookan-style carvings are present on the interior posts and the back wall. Photo taken in 2013. Photo by Jon Daehnke.

As I got more interested and wanted more information, there just seemed to be this rather big vacuum. There were a couple of people who had the information but who were reluctant to share it. If you can't get information from them, you'll find workarounds. We were just coming into the computer age where more and more information is available online. So, I started collecting photographs of artifacts, as many as I could find from wherever I could find them and just compiling them so I could study and look for repetitions in form, those things that make it the form. The Plankhouse kept me busy, but after the Plankhouse was done, I found the time to continue and I'd started a couple of things such as a bowl and a canoe model. I think I'd started on a comb by that time, using the information mostly from photographs that I had gathered. I'd also become acquainted with Verne Ray (anthropologist, author of "Lower Chinook Ethnographic Notes", Verne Ray 1952), and read the *Chinook Texts* (Boas 1894). Of course, there's a wealth of knowledge in Verne Ray.

But the stories, the Chinook stories, are really the key to the art. You can't do this art, it's difficult to do this art, without being tied directly into those stories, the songs.

So, that's what started it. And then the real spark after that which really lit the fire was an openness to my work by Cecily Quintana [Quintana Galleries, Portland]. I was out of a job because the Plankhouse was finished and I was like "Well, what am I going to do now?" I wanted to find a way to continue my work, continue to do something with the Chinook. There was a brief rumor of hope for Chinook federal recognition, and I thought a position with the tribe would maybe become available. But that evaporated. So, I approached Cecily. I had a couple of things underway, and maybe one piece finished. She was open to the idea and excited about it. I was skeptical because we had discussed before about how Northern Formline is such a powerful art form, especially in the gallery setting. It is really eye candy and is very attractive to those buyers and collectors. I wasn't sure how Chinook would stand up to that because the form is more simplified—I guess you could say more primitive or crude—comparatively. But she had me carve a number of pieces, and put on a solo show. The pieces sold almost right off and that encouraged that. That was the encouragement to move forward and continue pushing for the art (Figure 5a,b).

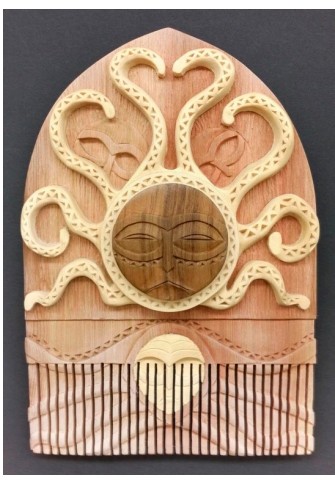

(**a**)

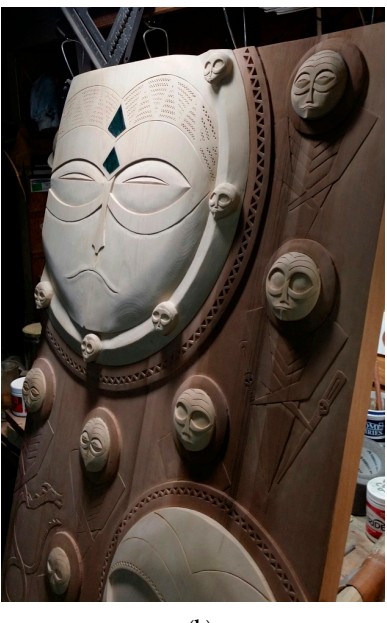

(**b**)

**Figure 5.** Octopus comb (**a**) and wood panel (**b**) by Greg Robinson, both in private collections. Photos courtesy of Greg Robinson.

**JD:** We started to talk a little bit about this already, but I'd like to get back to some of the influences or guides that you use to create and develop your work. And maybe put that within the context of what's happening with Chinookan art from the time of contact and into the 19th and 20th centuries, during that period where...

**GR:** Yeah, the void. What I call the void. So, it's a complicated set of events. You have diseases coming up out of California, diseases coming from the Spanish even earlier, before direct contact. There are waves of disruption. And every time you have a disruption...I always use the example of a canoe maker who's teaching who knows how many students and family members how to carve these canoes. So, let's say there are ten people involved in carving that canoe, maybe two old-timers and maybe a middle-aged guy; and they're all contributing to teaching the younger kids, some who are on their first canoe, some who are young enough to just be watching, some who are actually participating. There are going to be taboos that are associated with all that process because there are taboos associated with every second of everyday life. So, it's not just a simple picture of these guys all gathering around this canoe. It's quite a complicated thing happening.

So here comes a wave of—take your pick—influenza, smallpox, malaria. Half of those people die, maybe one of the old timers, maybe a couple of the old timers, a couple of the kids. And so, you look at the gap that now creates, both in the knowledge of the taboos that are supposed to be followed and the knowledge, the physical knowledge, of how this canoe is supposed to be carved. And of course, whatever stories are tied into that, whatever songs are being practiced during that. But they've forged forward with this now fragmented knowledge. Maybe they're filling in gaps, but it's not the same as it was before. So here comes the next wave and you lose another percentage of those people. And there are people stepping in to take those places, but now they're not privy to all the information that was handed out to the pre-disease people. So, for every step forward you get this greater dilution and fragmentation. Now it's a crap shoot as to who's survived and how much knowledge. So that's an example of that lost, fragmented, and diluted culture that transpired.

I think from about the early 1900s to, I'd say, the 1980s maybe, there's not much going on. There are a few families producing their variations of a little bit of art, some paintings. There's a few of them that knew about the Shoalwater carving [wood carving of a human figure, presently at the Shoalwater Bay Museum, Shoalwater Indian Tribe], which I think you're familiar with, the real popular one. That was probably one of the pieces that influenced a lot of people in that void time. I don't know when that piece was produced. I think it was produced fairly late; I don't think it's particularly old.

Also, you had Chinookan family members that maybe had some art pieces, some old pieces. So, they were aware of kind of a shadow of what was going on. But nobody, to my knowledge, is producing full-on hardcore Chinookan Columbia River art during that void. Once those last elders passed in the early 1900s, like Ch'isht and Cultee, maybe within that next generation, they may have passed some information down to those direct relatives underneath them. But I think you start to see a rapid disintegration of that knowledge being passed. Some songs survived, some dances survived. Some ceremonial knowledge of ceremonies survived. But not much in the way of produced art. What was being produced were generally copies of what they'd seen, what few pieces were available to be seen. That's not to say there wasn't some information. The Oregon Historical Society had some publications that had quite a bit of artifact photos and was probably one of the earliest available sources to look at for the art form, other than what was already in museums. And then eventually Bill Holm produces that laser disc that has quite a bit of information on it. That was an early influence for me. But truthfully, there wasn't a lot of compiled information. It was all scattered. And also, I would just say there's more in private collections still than has ever been compiled in any kind of a recorded collection, institutional collection. So, I started watching auction sites and found a plethora of unseen, valuable additions to the stuff that had previously been published.

**JD:** Were these local auction sites, or...?

**GR:** Auction sites just sort of throughout the world. Including Sotheby's and all the big auction houses. There's a long history of contact on the Columbia River. So, we have people starting to collect and pick up stuff at that early time period. And then, that tradition of collecting and that pot-hunting kind of mentality continues, well, to some degree even into today's world. But it was very much a popular thing from the 1930s throughout the 1950s and early 1960s even, where it was common to just go out for the weekend and dig for artifacts. So, there's just tons of material that's been collected and hoarded away in cardboard boxes. And eventually, they find their way to whoever gets them when they are handed down when somebody passes away. So, a lot of that stuff people collected and accumulated in the 1940s and 1950s gets passed down to the kids of those people doing the accumulating. And they don't necessarily know what to do with them or want them, so they end up at an auction site. So, auctions are one of the sources I keep an eye on to see pieces I haven't seen before. I've seen most of the pieces that are in public collections. But if you watch the auction sites, you'll see pieces that you haven't, that nobody, or very few people have ever seen. Yes, those are still being sold. Some of them are funerary items. And some are really unique and exotic pieces that just add to the amount of information that is available to the art form today. There's a lot of information available now in just the last 20 years. And so, that really kind of opened the field for me, because in seeing, finally in seeing enough photos, you realize how much flexibility within the art form there is.

So, we had the void, and then about the earliest person I'm aware of was Tony Johnson [current Chair of the Chinook Indian Nation] who started putting some of the pieces together early on as he was investigating the language. And he had exposure to some elders. He knew about Verne Ray, of course, and the *Chinook Texts* before I did. I think some of the most early modern pieces you see being produced are by Tony.

**JD:** So that brings us to revitalization. There's been this tremendous revitalization and growing interest in Chinookan art in the last few years.

**GR:** Really just in the last ten years, yeah.

**JD:** What do you think has fueled that?

**GR:** Well, in general, there's just a whole Native resurgence in contemporary art and folks saying "We're not just antiques. We're in the modern world and we're producing modern art. Why aren't you taking a look at that?" I don't know what catalyzed it, or what got it started…probably just a new generation of contemporary Native artists that are voicing their opinions through the power of social media for exposure. It's kind of hard to ignore these young rebellious activists and so, why not embrace them?

For Chinook, in specific, there are several things. I would say social media is a huge driver of that, just because so much information is available at a snap of the finger. You'll send, you know, a picture from your phone of your work. I got on Facebook pretty late. But it became apparent really quickly for me and my art, that it was going to be a vital tool because I could show my work and process. And all the friends that wanted to be on my friends list can see that. I can focus on art-related people.

The other huge spark, not just for us but for everybody, is Canoe Journeys. Because all of a sudden, you're put in this very emotional experience. "Life-changing" is something you'll hear constantly from people who have gone on that. I encouraged the Tribe to go on that for years before we finally took the plunge and got a canoe made and took the journey and people came back and they're changed. And so now, all of a sudden, during protocols and all those different stages of protocol, you've got to show off what you do (for more on Canoe Journeys and protocols, see Daehnke 2019; Cushman et al. 2021). Who you are, what's special about your Tribe, and what you can do. All you're going to do is show off. Now, you have this really strong interest in producing your own work to show who you are. You're not showing some adopted art form because you're up against…well, you're in there with Alaskan and Canadian natives! And if you're showing some weird form of formline, they're not receptive to that. They're not appreciative of that. And so, that's been a huge fire built under these tribal members to do their own work. So, in that sense, you start seeing this push of formline slowly being worked back out. So yes, you

see a driving push to revitalize Chinookan art in that. And for Grand Ronde that was the spark. At that same time, I'm also really moving forward with my work, producing a lot of pictures for people to see. So, social media, and Canoe Journey. And I would say Canoe Journey primarily. Then once you start that fire, and people start producing work, carving classes start getting taught, and younger kids start wanting to participate. So, they then grow up seeing a proper art form. It just becomes what they know, what they identify as the starting point. So now you've planted the seeds for future generations to at least have a proper starting point, and not have to go backwards and get rid of all the other influences first. They can just start from that point and move forward.

Then, as time goes by and people are growing up producing more and more Chinookan art you end up just seeing greater interest. Because now there are more artists producing more work. There's more work available to go to galleries. And with public art, specifically, there's been a drive for this in Portland through social media, pressure in social media to acknowledge the people that are actually from the area. They get guilted into doing the right thing, which is to show the work that properly belongs here.

**JD:** Perhaps a form of land acknowledgment.

**GR:** Yeah. But that's still got a long way to go because there are still outside art forms, Northern Formline, being produced as public art in Portland. I think you and I talked previously about that Preston Singletary piece that was recently installed in the Pearl District in Portland ["Dancing Staff", Dianne Apartment Building]. So, there are still those pieces being installed, and to some degree, I think that is fine, it's a beautiful piece, and I think there's a place for that. But it would be nice if the people from the area actually recognize the work that is from here, this place. It's still a huge mystery to a lot of people.

But the Cathlapotle Plankhouse has been helpful with that recognition. One of the reasons I took that at times nightmarish job there, was knowing that it would become a field trip destination. So at least new generations of kids would know and could say "Oh, look, this is Chinookan art and these are the type of houses people lived in here." So, you see public art—with Chinookan form—has quadrupled in less than ten years. Maybe even more than that, maybe ten times. Because I don't think, I don't even know if there were any public Chinookan art pieces in Portland before that, were there? There were maybe some vague references here and there. I'm sure there must have been something, but I just can't think of any.

**JD:** What was your first public art piece?

**GR:** The first public art piece was a basalt carving at Parker's Landing Historical Park in Washougal, Washington. That's my first foray into public art. Greg Archuleta [Confederated Tribes of Grand Ronde] and I both worked on that piece. I used to brag about the fact that it was the tallest Chinookan stone piece, usurping the well-known historic one in the Portland Art Museum—the six-footer—by making mine eight feet. It's actually closer to ten with the piece on the top. I picked stone just because of the potential vandalism issues and its longevity and ended up putting, I don't know how many tons of stone in there altogether (Figure 6).

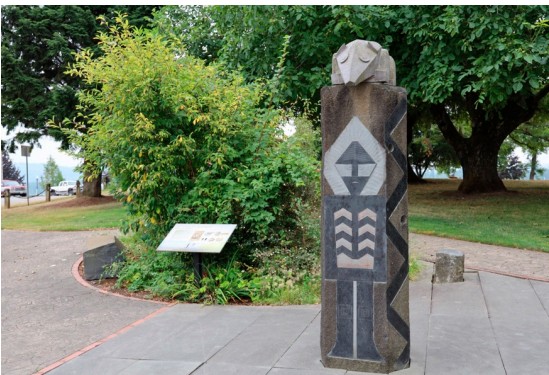

**Figure 6.** Chinookan Welcoming Figure (basalt). Chinook Plaza, Parker's Landing Historical Park, Washougal, Washington. Photo by Jon Daehnke.

**JD:** What year was that? Was that 2005, roughly?

**GR:** 2007-ish I think. 2006, 2007. The project just kept growing, and we ended up doing a lot of other stonework there. It spread out over there for a couple of years. That then led to the pieces that are currently on the new pedestrian bridge in Portland [Tilikum Crossing Bridge], which were originally slated for Multnomah Falls. Multnomah Falls got a grant, and I think that it was through Greg Archuleta that they became aware of my stone piece at Washougal and wanted to do something similar at Multnomah Falls. So, I agreed to take it on. The original idea was three pieces—two standing stones with a bronze circle in the center. I was excited about it. And I did it for cheap because of the millions of people who would take their photograph standing by my art. That was worth it to me. But long story short, they didn't actually have the authority to install the art there. And the guy that originally was very supportive of it left. Then their historical department got involved, as well as a whole set of governmental people, and then they said "No, it doesn't match the historical surroundings"! This got Grand Ronde involved again because it was a Grand Ronde grant—a Spirit Mountain Grant—that had helped fund it in the first place. I encouraged Grand Ronde to play hardball with them, and their lawyers got involved, and eventually, they decided to just look for a different home for it. Then eventually along comes the pedestrian bridge [Tilikum Crossing Bridge] and it ends up going there. So, that was my second big piece. And maybe those are the first couple of significant pieces that are in the Portland area (Figure 7a,b).

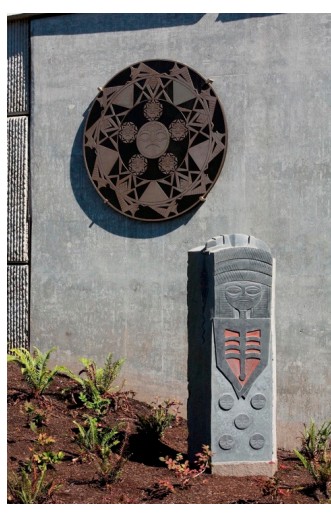

(**a**)

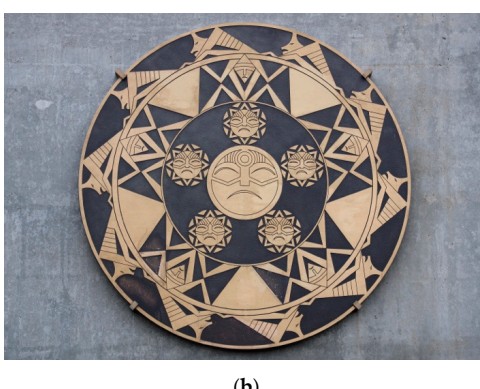

(**b**)

**Figure 7.** "We Have Always Lived Here" public artwork on the east side of the Tilikum Crossing Bridge, Portland, Oregon. The basalt figure (**a**) is a Tayi or headman. An additional basalt figure sits on the west side of the bridge (not shown). The bronze medallion (**b**) is Morning Star with her five children in the interior ring, and Coyote traveling the world and establishing rules for humans to bear on the outer ring. Photo (**a**) courtesy of Greg Robinson, photo (**b**) by Jon Daehnke.

**JD:** Had you been working with Greg Archuleta and the Grand Ronde for a while?

**GR:** Right around the time of the first stone, so 2007, I got involved with Lifeways [a cultural education program at the Confederated Tribes of Grand Ronde]. Greg Archuleta had started that program a couple of years before. So, that puts me in a classroom situation, which gives me even more drive to make sure I know what I'm doing. And so, I started openly sharing my work. If you see something online, you're welcome to use it. Change it up a little bit, preferably. But use it. All of those art elements don't belong to me. I can't claim rights to them. They're just elements I've taken from scattered things and put them together into something and made something out of it. So, they don't belong to me anyway. I've always been open in sharing that, and I was just hoping to get somebody the spark.

**JD:** How many people are actively carving now, at least those who have been coming out of some of that work with Grand Ronde?

**GR:** I'd say probably a dozen people. One of our success stories is Jordan Mercier, who didn't initially know much about Grand Ronde, though he's a tribal member and his dad was a tribal member. He started to come to the Lifeways classes, started learning some weaving and some art design, and he started singing. He's a good singer. And so, Greg was quick to take advantage of that because he was doing a lot of public work around the city, and that work involves singers. So, the singing led Jordan to get more involved. Through that connection, he eventually got on temporary pool, and then he moved from temporary pool to full-time employment, and then moved up into a top position in the cultural department at Grand Ronde. He carves and he sings and he does all the art, and now he teaches people to do it as well. So yeah, probably a dozen or so, mostly in Grand Ronde.

**JD:** It seems like there's a pretty substantial group of carvers and artists now.

**GR:** There's still a long way to go. I think that there should be 100 carvers. But, I would say there are half a dozen fairly prolific carvers, and then there are another half a dozen who just carve for themselves and do occasional pieces, and then a dozen or so that are just kind of dabbling in it. But yeah, it's more than when I started.

**JD:** Are you still teaching? Are you still working with Grand Ronde?

**GR:** Yes.

**JD:** How often is that?

**GR:** It's once a week. Greg [Archuleta] works full-time for the tribe now, so there are a few more interruptions to the schedule than there used to be. But as a general rule, it's every week. They're intertribal in focus. We try to steer people to their own cultures, trying to get away from the Pan-American Indian thing. We don't strictly enforce it, though. You see students from Grand Ronde who come in and want to do Northern Formline. We don't tell them no. But we do hope to encourage them to investigate their own culture. We have the materials, the photographs, the pictures, and the stories. All of that that they can lean on to learn. More and more, a component of this is the number of people that can speak the language. The Wawa [Chinuk Wawa] has just grown a hundred-fold.

**JD:** What's been the response—in the Portland region in specific, but also a little more broadly as well—to your work and the Chinookan style?

**GR:** I think it's still difficult—especially as far as gallery work goes—it's still a little difficult to sell once you get to Seattle and beyond. When Cecily Quintana initially retired, that left me without a local gallery. So, I approached Steinbrueck Native Gallery in Seattle because I always originally intended to be in three galleries to follow the Chinookan triadic symbol thing. I kind of got into three galleries, but mostly the work was in Quintana's and Steinbrueck's. She [Steinbrueck] was receptive, although a bit skeptical. She of course deals in almost all standard formline work. But my pieces started selling up there, so she became more receptive. As Quintana retired, it became a vital source for gallery sales. But she [Steinbrueck] always felt people weren't sufficiently educated on it, they didn't know what they were looking at. But for collectors who wanted something different that they hadn't seen, Chinook provided a new thing that nobody had been collecting. There was

that avenue and that helped a lot. But I think Seattle is probably the border where you could really move a lot of Chinookan work.

**JD:** Are you continuing to get requests from the city, or from other local organizations for public work?

**GR:** Yeah. I'm not sure exactly when I became more on the radar for public art. I know that Phillip Hillaire has been a driving force for getting local Indigenous art represented in Portland. Also, I think my name just finally got enough recognition that when they were looking for local art it came up. Lillian Pitt also has been a big advocate for me. Lillian, of course, is kind of the matriarch we all look to, and somebody like her really helps to get you on their radar. Of course, once you start doing some public art, you just get on that list of people that they approach. So yes, the city of Portland, quite a few organizations, some connected through Quintana, because Cecily, of course, knows a boatload of people in this area. She was involved in the collection of combs that I did for the Portland Building. That was a significant thing. The new office of the Bonneville Power Association did commission some pieces also. And I got approached by the City of Vancouver, Washington, specifically by the mayor because she travels abroad and likes to take gifts from the local area. So, the comb I just finished went to Japan as a gift. I made sure it was a good piece of cedar because they have a reverence for cedar in Japan. So, a few pieces overseas. She also gifted one of my paddles to Chuck Sams, the head of National Parks.

**JD:** And the power board that you did for the Japanese Garden exhibit, did that get moved?

**GR:** Yes, it got moved to outside of the Oregon Convention Center (Figure 8). We went out there to get our vaccines and I remember it still looked pretty good. I put a high-tech coating on it and it held up pretty well in the weather. But it won't last forever though. Those things have a life and just like us, as we get older and gray shows up, those pieces are expected to age out as well.

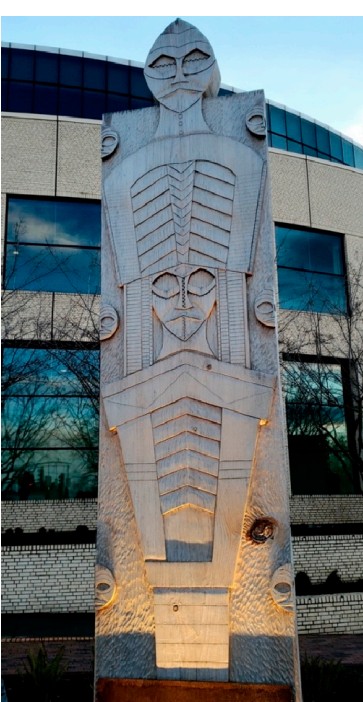

**Figure 8.** Chinookan "Power Board", formerly at the Japanese Gardens, now located at the Oregon Convention Center, Portland, Oregon. Photo courtesy of Greg Robinson.

**JD:** In addition to the carving, you paint as well. What's the connection between the carving and the painting? Do they drive each other?

GR: Well, originally, I was a pencil artist. I sold my first graphic piece in junior high or something to a teacher. But the paintings—what I call my "crazy paintings," my surrealistic work—those did not really start until 2018. At first, I thought, "you know what, I'm going to move into this and see because you never know until you try." I decided to see if anybody was interested in it because I was looking for something to supplement the wood carving, something that wasn't as hard or time-consuming. And also, it's just freedom, as opposed to the carving which is quite regimented. When I want relief from the carving, I could move into this. And I could wipe out a painting in an evening, or I might work on it off and on for a couple of weeks. There's no pressure to do anything. I could paint over it—and I have many times just painted it over and started it over. And then they started selling right off, just like the carvings. So, then I thought, "Okay, then I'm going to keep pursuing it" (Figure 9a,b).

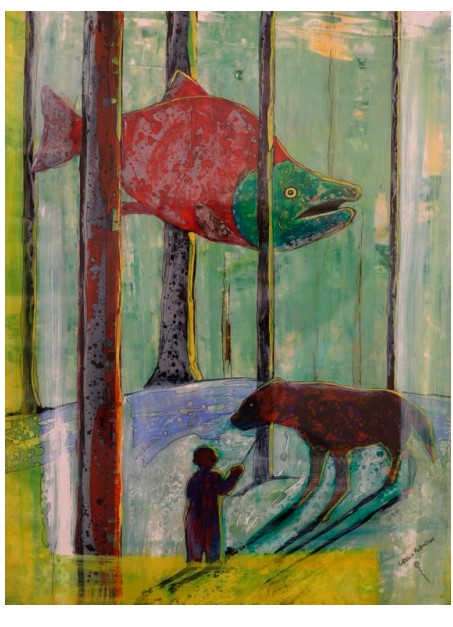

(**a**)

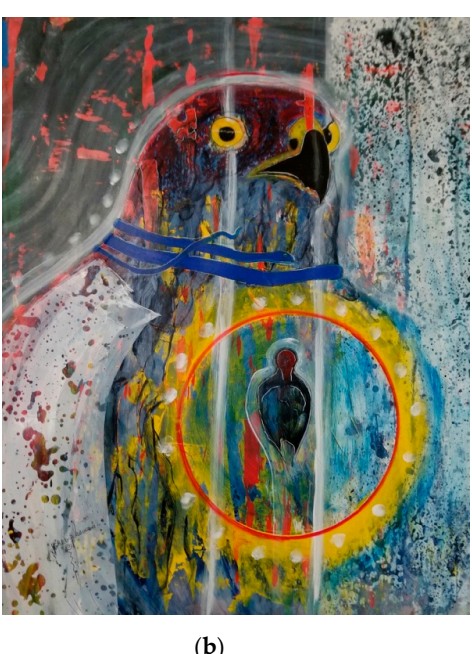

(**b**)

**Figure 9.** Two paintings by Greg Robinson, based on Chinookan stories. Photos courtesy of Greg Robinson.

**JD:** Is it the same sets of people buying both or is it two separate audiences?

**GR:** I think it attracted some people who wouldn't have bought the carvings. They're my attempt to illustrate that supernatural world, which if you and I were to travel to the land of the dead right now—we don't have the power to do that, of course—but if we had the power to do that, you would see those paintings. That blur and that craziness. Kind of like Dr. Strange's weird movies. Because we can't make sense of it because we're not supernatural or we're not dead. In the old stories, they talk about how Coyote specifically would go to the land of the dead to visit his sister Robin. He had the power to do it, but he couldn't stay there more than, I want to say seven days. There was a time limit. And if you didn't get out you were stuck there. But he saw just this jumbled craziness. People were skeletons and he would get enjoyment out of that, spooking them and startling them and their bones would all fall apart. He thought that was hysterical. They would go fishing and catch their leaves and pull their nets up and their nets were all full of holes. They'd pull up these leaves and twigs in their nets with holes and they'd be going, "Yeah, we got some nice fish here." Meanwhile, Coyote's going "What?". Same with elk hunting. They were all elk hunting and they have this drive and this guy shoots this chipmunk and he says, "Yeah, we've got this big elk!", and Coyote is going, "What? That's a chipmunk!" But he realized when he got back, all these things turned into what they were supposed to be. He realized that he was not seeing things as well as he could in that supernatural world. The paintings are a reflection of that type of experience, our inability to see the supernatural world in a way that makes total sense. The paintings are a totally non-traditional exploration of our stories. But they're still anchored in the old works, the old stories. So, no matter how contemporary you get, there's always a tether that anchors to something in the past.

**JD:** Well, we're reaching the end here, but before leaving I wanted to quickly revisit what you think is the central importance of having a revitalization of Chinookan art along the Columbia River, along the Willamette River, and in this area more broadly?

**GR:** I think that you can sum it up with one word, and that's education. That the public becomes educated and aware of the art that belongs here, belongs in this place. I think that is its most single importance. Of course, the art is beautiful and people appreciate beauty. But for me, it's most importantly about people recognizing it, and being able to say "Oh, that looks like the piece I saw at such and such. That must be a Columbia River piece." That type of place recognition. And that ties back to what we talked about before, having places like the Cathlapotle Plankhouse being a big destination for school children. Greg Archuleta and I did interpretation for the school groups there for a number of years, doing pigments, art, and different things. That education component, it ties into everything because as people become more aware, those people grow up to be collectors or buyers or they are involved in art committees that choose art for places. Having that education is important when you're coming off "the void" where there's been no awareness or very little awareness, other than what's scattered in some history books.

## 3. Conclusions

In our conversations about Chinookan art and its resurgence over the years, there are two broad themes that Greg consistently returned to. One of these themes, as just noted, is the central importance of educating the general public on the Chinookan art form and its unique status in the Pacific Northwest. That education process includes not only creating a greater awareness of the form itself and its distinctive style, but also the very central place-based nature of the art form; a recognition that this is the proper form of art for the Lower Columbia River, a form created and practiced by the people who have lived there for millennia and who still reside there, and create art there, today. Chinookan Art is not without place: it is integrally attached to and shaped by the landscape itself and the people of that landscape. The second theme emphasizes the necessity of teaching the art form to Chinookan youth. Furthermore, teaching them that this is not just about making art, but is also integrally attached to cultural production more broadly, including connection to place, language, stories, protocols, and ultimately Indigenous identity itself. It serves as a source

of pride, resilience, and resistance. His hope is that creating art provides that further spark, and that "once the fire starts, then there is no stopping it." And where there were once generations who never saw a landscape with Chinookan art, there will now be generations who will never know a landscape without it.

**Author Contributions:** Both authors (J.D.D. and G.A.R.) shared equally in the creation, development, and content of the article. All authors have read and agreed to the published version of the manuscript.

**Funding:** This project was supported in part by funds from the Faculty Research Grants program at the University of California, Santa Cruz.

**Data Availability Statement:** Data is contained within the article.

**Conflicts of Interest:** The authors declare no conflict of interest.

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
