# Peer review of "“Once the Fire Starts Then There Is No Stopping It”: The Revitalization of Chinookan Art in the 21st Century, Conversations with Greg A. Robinson"

_arts, 2023_

Round 1

Reviewer 1 Report

This well-written article centers on interviews with Chinookan artist Greg A. Robinson to offer, as the title suggests, the artist's own account of the revitalization of Chinookan art in the 21st c. Historically, Chinookan art has been neglected in Northwest Coast art studies, so this article makes an important contribution to the field by sharing the words of one of the key artists in the revitalization movement. I have only a few suggestions for minor revisions the author might consider:

--the article launches with a discussion of the 2019 Forest of Dreams exhibition in Portland, which features both Chinookan and Ainu arts. Given the importance of this opening statement, I thought the Ainu would figure more prominently in the article as a whole, but the article ended up focusing on Chinookan art and Robinson in particular.  While the author makes clear that the Forest of Dreams exhibition is an example of Chinookan art repopulating the landscape in Portland--after many years of northern NWC arts dominating the scene--I found the discussion of Ainu art in the opening paragraph a bit of a red herring, since it does not reappear anywhere else in the essay. The author might consider whether to bring the exhibition discussion full circle and at least mention the Forest of Dreams exhibition in the conclusion, or whether to edit the opening paragraph slightly to signal that the article's main focus is on Chinookan art, not this exhibition of Chinookan and Ainu art.

--p. 3 last line of paragraph: The author indicates that "Chinookan art differs considerably from the more widely recognized [northern NWC art]" but ends the sentence without giving any examples or details of these differences. I I was hungry for a bit of a summary here on some of the distinct features of Chinookan art, or at least a promise that the article would discuss these differences in more detail later (as Robinson's interviews do).

--p. 5 related note to above: In the interview, when Robinson is speaking about the distinct motifs of Chinookan art (like 3s and 5s, negative triangles but positive zigzags), I was hungry for figure references that would allow me to see these motifs in the art. Could the author insert brackets to "see, for example," one of the figures already included in the article that would showcase these motifs? I see this article as an opportunity to teach more people about the under-recognized Chinookan art form, but to do so we need more explicit call-outs of these motifs and figures to see them in action.

-p. 6 line 187 clarify date of Lewis & Clark Bicenntenial (not everyone knows)

--Fig. 4 add date

--p. 7 line 218 clarify with brackets the identity of Verne Raye [an Elder?]

--p. 9 line 272 clarify Shoalwater carving--sounds famous but those of us outside Chinookan art don't necessarily know what it is--maybe a footnote would work well here if info can't be contained in a bracket

--conclusion: anything to tie back to Forest of Dreams exhibition? that opening seems a little bit of a red herring if it doesn't factor into conclusion as well

Author Response

Thank you for your careful and helpful review of the article. We appreciate the time you took to review this and the thoughtful comments that you provided. We have modified the article to take them into account.

  1. We reworked the opening paragraph to focus primarily on Chinook involvement in the Forest of Dreams exhibit, to make it clear that this is the primary emphasis of the work.
  2. We provided an indicator on page 3 that we would be discussing the differences/uniqueness of Chinookan art in upcoming sections of the paper.
  3. Beginning with page 5 we provided visual call-outs for when an artistic characteristic was being described (for instance "see Figs. 4, 7a, 7b), so that the reader could get an immediate visual reference for that characteristic.
  4. We added the dates for the Lewis and Clark Bicentennial.
  5. We added the date for Figure 4.
  6. We clarified who Verne Ray was, as well as adding the relevant work to the bibliography.
  7. We offered a brief description of the Shoalwater carving.
  8. Since we minimized the Ainu discussion in the introduction, we didn't feel the need to go back to the Forest of Dreams exhibit in the conclusion. We did, however, add a couple of additional sentences on the connection between Chinook art and Chinook place, as that is really a central idea behind the piece. 

Reviewer 2 Report

Interview is excellent and I am glad to see that the interviewer gives lots of space to the artist who is articulate, knowledgeable, and generous in speaking about other arists and his community. 

I have no edits to suggest. I enjoyed reading the interview.

Author Response

Thank you for your review of our article. We're glad that you liked the submission, and greatly appreciate your time and effort.

Reviewer 3 Report

Very interesting, and important, work.  However,  I had some questions as I read it that should be addressed to clarify the work.   Remember that most of your readers are unfamiliar with this work and the region. I found it very informative and it introduced me to a new field of art production.  My questions:

What is a POWERBOARD exactly?  What is the symbolism? I think they are beautiful but wanted to know more. 

More information about the Plankhouse - I never heard of that.

What is NORTHFORM carving and/or SHOALWATER carving?

You use the word PROTOCOLS - what protocols?  For whom?

Explanation of paintings is better than carving.

Place-based nature of art form - you could elaborate on that also.

Author Response

Thank you for your very thoughtful and helpful review of our article. We appreciate your time and effort with this, and have incorporated your suggestions where possible.

  1. We included a few sentences on "Powerboards" so that the reader could understand both what they are, and why they are considered so important to the Chinook.
  2. While we did not add an additional section on the Cathlapotle plankhouse, we did provide references for articles and other readings that provide more context to the story and controversies surrounding the plankhouse.
  3. We give a bit more of a description of the Shoalwater Carving, where it is currently located, etc.
  4. In terms of protocols, especially as they relate to Canoe Journeys, we provided references to additional readings that address those more in depth.
  5. We added a bit more wording in the conclusion to emphasize a bit more the place-based nature of Chinook Art.